# Analysis of Ultrasonic Machining Characteristics under Dynamic Load

**DOI:** 10.3390/s22218576

**Published:** 2022-11-07

**Authors:** Zhangping Chen, Xinghong Zhao, Shixing Chen, Honghuan Chen, Pengfei Ni, Fan Zhang

**Affiliations:** 1School of Automation, Hangzhou Dianzi University, Hangzhou 310018, China; 2Zhejiang Best Optoelectronic Co., Ltd., Jiaxing 314009, China; 3Zhejiang Jiakang Electronics Co., Ltd., Jiaxing 314001, China

**Keywords:** piezoelectric transducer, impedance characteristics, equivalent circuit theory, ultrasonic

## Abstract

This research focuses on the load characteristics of piezoelectric transducers in the process of longitudinal vibration ultrasonic welding. We are primarily interested in the impedance characteristics of the piezoelectric transducer during loading, which is studied by leveraging the equivalent circuit theory of piezoelectric transducers. Specifically, we propose a cross-value mapping method. This method can well map the load change in ultrasonic welding to the impedance change, aiming to obtain an equivalent model of impedance and load. The least-squares strategy is used for parameter identification during data fitting. Extensive simulations and physical experiments are conducted to verify the proposed model. As a result, we can empirically find that the result from our model agrees with the impedance characteristics from the real-life data measured by the impedance meter, indicating its potential for real practice in controller research and transducer design.

## 1. Introduction

Ultrasound is used in various industrial, agricultural, and medical applications, such as ultrasonic welding, imaging, and detecting [1,2,3,4,5]. With the international call for energy conservation and emission reduction, the prospects for electric vehicles are very good. The efficient and stable welding of batteries and wiring harnesses is an important guarantee for the normal operation of electric vehicles [6]. Conventional welding techniques mainly include: fusion welding, U-shaped terminal crimping, laser welding. In the battery connection process of electric vehicles, according to the technical requirements of welding, the most-used type is still laser welding [7]. However, the welding characteristics of laser welding materials with high reflectivity and high thermal conductivity will be changed by laser welding, resulting in a decrease in welding quality [8]. With the development of ultrasonic technology, the application of ultrasonics to the field of welding can achieve low resistance, high efficiency and a high-quality welding effect [9]. Ultrasonic welding is a kind of ultrasonic vibration-assisted machining method. The biggest feature of ultrasonic vibration-assisted machining is that uses a piezoelectric transducer for converting high-frequency electrical energy into high-frequency vibration energy, which can realize the related operation of the workpiece [10,11,12]. Through the above analysis, we can know that in the batteries and wiring harness welding of electric vehicles, ultrasonic welding technology has more advantages than conventional welding technology.

In the ultrasonic welding process, the stability of the amplitude of the welding head has a great impact on the quality of the welding, and the stability of the amplitude needs to be achieved by a controller with a good control effect. However, many previous studies on the transducer controller were conducted under no-load conditions and analyzed by establishing a model; it can only reflect part of the system’s performance [13]. After the load increases, the original no-load model does not reflect the actual situation of the system well, and the stability of the designed controller is poor. When ultrasonic-assisted vibration is subjected to a force load, the problems of system detuning and amplitude attenuation will occur [14]. This leads to the wear of the transducer tool head, which has a significant impact on ultrasonic welding, so it is necessary to study the load characteristics of the transducer.

In view of the uncertainty and randomness of the load characteristics of piezoelectric transducers in the ultrasonic welding process, we cannot grasp the dynamic characteristics of the ultrasonic welding process very well, and the quality of the workpiece cannot be guaranteed. In order to achieve a more stable and reliable welding process, it is very necessary to study the load conditions in the ultrasonic welding system. Aiming to achieve this research purpose, we need to conduct in-depth research on the electrical properties of piezoelectric transducers under load.

### 1.1. The Current State of the Research

Presently, domestic and foreign scholars have conducted some research on the load characteristics of piezoelectric transducers. Based on the equivalent circuit theory, the effect of liquid and solid loads on the load characteristics of the transducer is investigated. Although this is conducted to obtain the influence of load geometry on resonance frequency, it only provides a theoretical basis for the optimal design of a piezoelectric transducer without providing support for optimizing the transducer controller [15]. Furthermore, some scholars have conducted a qualitative study on the loading characteristics of the transducer. Based on the PSpice loss model of the piezoelectric transducer, the load characteristics of the piezoelectric transducer and the impedance analysis are performed from the perspective of the time and frequency domain [16]. The relationship between the electrical load and resonance frequency of the transducer is studied by establishing an equivalent circuit under electrical load. And the range of resonance frequency and amplitude are observed to change when the electrical load varies [17]. A dynamic impedance model with external force is established based on the electromechanical equivalence method. The frequency and impedance characteristics of the transducer under no load and external force, respectively, are then realized by transfer function [18]. The abovementioned scholars only qualitatively analyzed load characteristics when applying different loads on the piezoelectric transducer without further quantitative research. By developing an equivalent circuit model of a circular biomorphic ultrasonic transducer, the impedance and frequency under air and water loading conditions are investigated to better analyze the dynamic characteristics of the transducer processing [19]. They Designed a cascaded transducer consisting of three sets of sandwich-type piezoelectric ceramics connected in series and analyzed the relationship between several characteristic parameters under fundamental and second harmonic frequencies and the load. However, it does not investigate the existence of some correlation between load characteristic parameters and impedance [20]. Based on the one-dimensional ultrasonic vibration system, the influence of load on acoustic system characteristics of ultrasonic machining is studied by three different load modes [21]: through establishing a combined impedance model for ultrasonic transducers to predict frequency, resistance, and conductivity under different loads. This plays a vital role in the relevant applications of ultrasonic transducers. Nevertheless, specific impedance characteristics (capacitive, inductive, and resistive) of transducers when loads are applied have not been studied [22]. Based on the developed dynamic model, the effect of the thermo-mechanical load on the characteristics of the ultrasonic vibration system was researched, and the amplitude and frequency change is determined in a similar trend with the change in load. However, information about the relationship between different loads and impedance in actual operating conditions was not obtained [23]. A block diagram method is proposed to analyze the dynamic characteristics of the piezoelectric transducers. The influence of a force and current input on the frequency response of the transducer is studied, and the frequency of the transducer can be predicted accordingly [24]. By establishing the dynamic models corresponding to different forms, such as pure resistance load and inductance load, respectively, the frequency and amplitude output characteristics of the piezoelectric transducer under different loads are analyzed [25].

Some scholars have studied the no-load characteristics of piezoelectric transducers. For the load characteristics of the piezoelectric transducer, other scholars only conduct a qualitative analysis, which cannot provide effective guidance for ultrasonic welding amplitude control and frequency tracking. In this paper, we analyze the electrical characteristics of the piezoelectric transducer based on the electromechanical equivalent model. In view of the fact that the load direction in the ultrasonic welding process is mainly along longitude, which can be simulated by applying a longitudinal load to the front end of the tool head. By analyzing the characteristic of the loading experiment and the front cover plate radiated acoustic ($Z_{ft}$), we found that the $Z_{ft}$ is both capacitive and resistive. The cross-value mapping method can map the change of load to the change of impedance and establish an equivalent model both the load and impedance so that we can better grasp the dynamic load characteristics of the Piezoelectric transducer.

### 1.2. Contribution and Basic Organization of This Paper

The main contributions of this work are as follows.

(1) Aiming at the uncertainty of the load in the ultrasonic welding process, in order to better grasp the welding mechanism, a cross-value mapping method is proposed to determine the impedance values corresponding to different loads.

(2) Fitting the data by the polynomial, the data include the real and imaginary parts of the impedance obtained from (1) and the load. The least-squares strategy is used for parameter identification during data fitting, aiming to obtain an optimal model.

(3) Verifying the proposed method by extensive simulations and experiments, the result of our model agrees with the data from the impedance analyzer.

This article is divided into five sections. Section 1 describes the current status of domestic and international research. In Section 2, the electromechanical model of the piezoelectric transducer is presented. Section 3 describes building the experimental platform and using the cross-value mapping method to determine the impedance, and fit the data. Section 4 verifies the obtained results by extensive experiments. Finally, the conclusions are presented in Section 5.

Throughout the paper, notations used are standard, as shown in Table 1.

## 2. Equivalent Model of Piezoelectric Transducer

The ultrasonic welding system consists of a 19.3 kHz longitudinal vibration piezoelectric transducer, a welding head, and other parts. The piezoelectric transducer is composed of three parts, including a metal back cover, piezoelectric ceramic stacks, and metal front cover [26]. Although there are many methods to analyze the load characteristics of the piezoelectric transducer, such as a finite element, equivalent circuit method, and transfer matrices. By comparing and analyzing the advantages and disadvantages of different methods, this research uses the equivalent circuit method to analyze the relationship between the load and impedance of the piezoelectric transducer [27,28]. Furthermore, we integrate the equivalent models of metal front and rear covers and piezoelectric ceramic stacks. In this study, due to the fact that the change of vibration shape has little effect on the process of establishing the equivalent model of load and impedance, we ignore the influence of the load on the shape of the vibration. The electromechanical equivalent model of the piezoelectric transducer is obtained as shown in Figure 1.

From Figure 1, $Z_{11}$, $Z_{12}$, and $Z_{13}$ are the characteristic mechanical impedance of the back cover. The impedance at the connection between the front and back covers and piezoelectric ceramic, respectively, are represented by $Z_{m2}$, $Z_{m1}$; The $Z_{1p^{\prime }}$, $Z_{2p^{\prime }}$ and $Z_{3p^{\prime }}$ and denote the impedance of the piezoelectric ceramic; The $Z_{21}$, $Z_{22}$ and $Z_{23}$ are the characteristic mechanical impedance of the front cover. Where $Z_{ft}$ represents the radiated acoustic impedance of the front cover, $Z_{f}$ represents the radiated acoustic impedance of the metal back cover in Figure 1. The metal back cover does not directly contact the load; rather, its contact medium is air or the material with less acoustic impedance. So, the $Z_{f}$ can be approximated as a short circuit treatment, namely $Z_{f}$ = 0. The series-parallel relationship of the components in Figure 1 can be calculated according to the Mason equivalent circuit theory. We have
(1)$$\left\{\begin{array}{c} Z_{l}=\frac{Z_{11}Z_{13}}{Z_{11}+Z_{13}}+Z_{12}+R_{m1}+Z_{1p^{\prime }}\\ Z_{r}=\frac{\left(Z_{22}+Z_{ft}\right)Z_{23}}{Z_{22}+Z_{ft}+Z_{23}}+Z_{21}+R_{m2}+Z_{2p^{\prime }}\\ Z_{s}=\frac{Z_{l}Z_{r}}{Z_{l}+Z_{r}}+Z_{3p^{\prime }}\\ Z_{e}=\frac{1}{n^{2}}Z_{s} \end{array}\right.$$
where $Z_{l}$ denotes the total impedance of the metal back cover obtained by connecting $Z_{11}$ and $Z_{13}$ in parallel and then in series with $Z_{12}$, $Z_{m1}$ and $Z_{1p^{\prime }}$ in that order. The total impedance of the front cover $Z_{r}$ is the result of $Z_{22}$ in series with $Z_{ft}$, then in parallel with $Z_{23}$. Finally, the result cascade with $Z_{21}$, $Z_{m2}$ and $Z_{2p^{\prime }}$. $Z_{s}$ represents the total impedance of the metal rear cover plate $Z_{l}$ being connected in parallel with the total impedance of the metal front cover plate $Z_{r}$ and then in series with $Z_{3p^{\prime }}$. To simplify the calculation process, we define some parameters, and the simplification table is shown in Table 2.

The meaning of symbols in Table 2 are explained as follows.

$\rho_{*}$: Material density

$v_{*}$: Longitudinal sound velocity in materials

$S_{*}$: Cross-sectional area

$k_{*}$: Material longitudinal wave number

$l_{*}$: Length of material

*m*: Number of the ceramic chip.

We will assign 1, 2, *p* to ∗, substituting the variables $Z_{11}$, $Z_{12}$, $Z_{13}$, $Z_{21}$, $Z_{22}$, $Z_{23}$, $Z_{1p^{\prime }}$, $Z_{2p^{\prime }}$, $Z_{3p^{\prime }}$ in Equation (Equation 1) can be obtained through Table 2, that is
(2)$$\left\{\begin{array}{c} Z_{l}=R_{m1}+\left[T_{1}+T_{p}+\frac{T_{1}S_{1}}{T_{1}+S_{1}}\right]j\\ Z_{r}=R_{m2}+\frac{RS_{2}\left(T_{2}+S_{2}\right)-RT_{2}S_{2}}{R^{2}+{\left(T_{2}+S_{2}\right)}^{2}}+\left[T_{2}+T_{p}+\frac{T_{2}S_{2}\left(T_{2}+S_{2}\right)+R^{2}S_{2}}{{\left(S_{2}+T_{2}\right)}^{2}+R^{2}}\right]j. \end{array}\right.$$

To avoid expression redundancy, replace the real and imaginary parts of $Z_{l}$ and $Z_{r}$ with $A_{11}$, $A_{22}$, $A_{33}$, and $A_{44}$, then
(3)$$\left\{\begin{array}{c} A_{11}=R_{m1}\\ A_{22}=T_{1}+T_{P}+\frac{T_{1}S_{1}}{T_{1}+S_{1}}\\ A_{33}=R_{m2}+\frac{RS_{2}\left(T_{2}+S_{2}\right)-RT_{2}S_{2}}{R^{2}+{\left(T_{2}+S_{2}\right)}^{2}}\\ A_{44}=T_{2}+T_{p}+\frac{T_{2}S_{2}\left(T_{2}+S_{2}\right)+R^{2}S_{2}}{{\left(S_{2}+T_{2}\right)}^{2}+R^{2}}. \end{array}\right.$$

Finally, the expression of $Z_{e}$ can be derived as
(4)$$\left\{\begin{array}{c} Z_{e}=\frac{A_{11}A_{33}^{2}+A_{33}A_{22}^{2}+A_{11}A_{44}^{2}+A_{33}A_{11}^{2}}{\left[{\left(A_{33}+A_{11}\right)}^{2}+{\left(A_{44}+A_{22}\right)}^{2}\right]n^{2}}+[\frac{S_{p}}{n^{2}}\\ +\frac{A_{22}A_{44}^{2}+A_{44}A_{22}^{2}+A_{44}A_{11}^{2}+A_{22}A_{33}^{2}}{{\left(A_{33}+A_{11}\right)}^{2}+{\left(A_{44}+A_{22}\right)}^{2}}\frac{1}{n^{2}}]j. \end{array}\right.$$

According to the Equation (Equation 5), the real ($R_{e}$) and imaginary ($X_{e}$) parts can be expressed respectively, that is
(5)$$\left\{\begin{array}{c} R_{e}=\frac{A_{11}A_{33}^{2}+A_{33}A_{22}^{2}+A_{11}A_{44}^{2}+A_{33}A_{11}^{2}}{\left[{\left(A_{33}+A_{11}\right)}^{2}+{\left(A_{44}+A_{22}\right)}^{2}\right]n^{2}}\\ X_{e}=\frac{S_{p}}{n^{2}}+\frac{A_{22}A_{44}^{2}+A_{44}A_{22}^{2}+A_{44}A_{11}^{2}+A_{22}A_{33}^{2}}{{\left(A_{33}+A_{11}\right)}^{2}+{\left(A_{44}+A_{22}\right)}^{2}}\frac{1}{n^{2}}. \end{array}\right.$$

The input impedance is expressed as
(6)$$Z_{i}=\frac{Z_{e}}{j\omega C_{0}Z_{e}+1}.$$

From Equation (Equation 6), we can see that the corresponding impedance will change when the load change during the ultrasonic welding process and the characteristics of the piezoelectric transducer will be affected. In order to better grasp the dynamic characteristics of machining and ensure the quality of machining, we can study the load and input impedance relationship.

## 3. Equivalent Model of the Piezoelectric Transducer about the Load and Impedance

In this paper, the equivalence model between the load and impedance of the piezoelectric transducer is studied during the longitudinal vibration of ultrasonic welding. The main research idea is shown in Figure 2.

First, we establish an electromechanical equivalent model of the piezoelectric transducer under load conditions and analyze the electrical characteristics of the transducer. At the same time, by analyzing the characteristic of the loading experiment and the front cover plate radiated acoustic impedance ($Z_{ft}$), we found that $Z_{ft}$ is both capacitive and resistive. Next, the electrical parameters ($f_{s}^{s},f_{1}^{s},R^{s}$) corresponding to the $Z_{ft}$ of n groups are simulated, and make the corresponding 3D diagram ($f_{s}^{s}-Z_{ft},f_{1}^{s}-Z_{ft},R^{s}-Z_{ft}$), respectively. The electrical parameters ($f_{s}^{e},f_{1}^{e},R^{e}$) are measured under m groups load in step 2.2. The electrical parameters measured by the experiment are cross-sectioned to the corresponding simulation analysis diagram in step 2.1, and the intersection lines are mapped in step 3. Making an inscribed circle and a circumcircle, respectively, to find the impedance corresponding to the selected load from m groups. Finally, the impedance and load are fitted by polynomials, the least squares method is used to find the optimal parameters. By comparing the three methods (inscribed circle, circumcircle, and average of inscribed circle and circumcircle), the best mapping relationship between impedance and load is obtained.

### 3.1. System Platform Introduction

In this experiment, the equipment involves a fixed device, a PV520A type impedance analyzer which is made by Beijing Band Ear Co. (Beijing, China), a piezoelectric transducer, weights, signal generator (AFG3101, Tektronix, Beaverton, OR, USA), oscilloscope (RTB2004, ROHDE & SCHWARZ, Muenchen, Germany), power amplifier (ATA-4011, Aigtek, Xi’an, China) and computer. The experiment platform is shown in Figure 3.

The experiment operations include fixing the transducer using a fixed device to ensure that the cross-section of the front cover can uniformly stress when force is applied. Next, a round sheet of the same material is placed on the end of the front cover, and the PV520A type impedance analyzer is connected to a piezoelectric transducer. Gradually, weights are added to the sheet, and at the same time, a computer is used to collect data when applying different weight.

There are many types of piezoelectric transducers available on the market. After consulting the relevant information, the common materials for the front cover of the piezoelectric transducer are aluminum, steel, and titanium alloy. Piezoelectric materials commonly utilized are quartz crystal, barium titanate, and lead zirconate titanate, the material used for the back cover is steel. The relevant parameters information of the piezoelectric transducer used in this research is shown in Table 3.

Since piezoelectric ceramics have a piezoelectric effect, the impedance characteristics of piezoelectric ceramics change when the frequency of the excitation signal at both ends of the piezoelectric ceramic sheet is changed. The impedance characteristics of the piezoelectric transducer are analyzed by means of a PV520A type impedance analyzer, and the curve derived is shown in Figure 4, where Figure 4a describes the admittance circle with ($1/2R_{e}$, $C_{0}\omega \rho$) as the center and $1/2R_{e}$ as the radius. Figure 4b presents the amplitude-phase curves, in which the red curve represents the amplitude curve and blue denotes the phase-frequency curve.

### 3.2. Loading Characteristic Analysis of Piezoelectric Transducer

**Remark 1.** 
*By comparing the electrical parameters obtained from the actual loading experiment and the $Z_{ft}$ with different impedance characteristics, we see that the impedance characteristic of the $Z_{ft}$ is both capacitive and resistive between $f_{s}$ and $f_{1}$.*


Figure 5 and Figure 6 show the impedance characteristic curve of the piezoelectric transducer when loading with different magnitudes of weight. From Figure 5, we can see that as the load increases, the radius of the admittance circle gradually decreases. At the same time, according to the amplitude curve graph, it can be observed that when the load increases, the amplitude decreases, and the amplitude curve has a right shift, as shown in Figure 6.

Based on the equivalent circuit model, the electrical parameters of the piezoelectric transducer corresponding to the radiated acoustic impedance of the front cover with different impedance characteristics are calculated separately. It is assumed that the range of radiated acoustic impedance of the front cover change during loading is resistive (0 to 1000), inductive (1+1i, 1+11i, 1+21i, …, 991+991i), and capacitive (1−1i, 1−11i, 1−21i, …, 991−991i), respectively. According to the component parameters information of the piezoelectric transducer in Table 2 and Equations (Equation 1)–(Equation 5). Matlab is used to calculate the amplitude-phase characteristics of the piezoelectric transducer under different $Z_{ft}$. The amplitude-phase curves of the piezoelectric transducer are shown in Figure 7, Figure 8 and Figure 9.

From Figure 7, we can see that when $Z_{ft}$ is larger, the peak value of curve decreases, and the lower proportion of phase curve in inductive load region. When $Z_{ft}$ is increased to 750Ω, the piezoelectric transducer exhibits capacitive characteristics over the entire sweep range, such as half power point, which does not exist. It can be seen from Figure 8, with the gradual increase in $Z_{ft}$, the amplitude curve decreased gradually, and the amplitude-phase curves move slightly to the right, with capacitive characteristics more than the entire frequency sweep range. Simultaneously, according to Figure 9, its peak of amplitude curve gradually decreases with the increase of $Z_{ft}$.

Comparing Figure 5, Figure 6, Figure 7, Figure 8 and Figure 9, we can find that the impedance characteristic curves of the loading experiment and the radiated acoustic impedance of the front cover plate are consistent with the trend of change under the capacitive load. This will provide a theoretical basis for finding the impedance corresponding to different loads.

### 3.3. Cross-Value Mapping Method Map the Load Change in Ultrasonic to the Impedance Change

**Remark 2.** 
*The impedance corresponding to different loads is creatively found by the cross-value mapping method.*


Additionally, we study the relationship between the load and impedance of the piezoelectric transducer, and a cross-value mapping method is proposed creatively. This is described in detail as follows: Through MATLAB simulation analysis, we can obtain a three-dimensional (3D) diagram of the load characteristic parameters. Then, selecting one of the multiple sets of characteristic data, which is measured in this experiment, we can make a horizontal plane in the 3D coordinate system for the selected data, and perform a horizontal cross-section with the 3D drawing of the drawn load characteristic parameters. Finally, the intersection curves are mapped to a two-dimensional plane, which consists of the x-axis (real part of the impedance) and the y-axis (imaginary part of the impedance). Furthermore, several intersection points are determined from map curves, making a maximum inscribed circle to the intersection point, the center of this circle denotes the impedance under this pressure load.

The most important parameters when analyzing the load characteristics of the piezoelectric transducer are frequency and resistance. By taking a comprehensive view, three load characteristic parameters of the resonant frequency $f_{s}$, half power point $f_{1}$, and dynamic resistance *R* are selected for this study. The range of impedance is 0 to a−bi (0≤a≤1000, 0≤b≤1000) for the research. To observe the relationship between impedance and load characteristic parameters ($f_{s}$, $f_{1}$, *R*) in the MATLAB simulation environment, the relevant information about front and back covers and piezoelectric ceramic crystal stacks in Table 2 is entered as basic information, Equations (Equation 1)–(Equation 5) are input as the calculation part. Based on the above information, the value of load characteristic parameters of the piezoelectric transducer in the selected impedance range can be calculated, and the 3D diagram corresponding to the values of characteristic parameters ($f_{s}$, $f_{1}$, *R*) can be formulated. The result plots are shown in Figure 10.

Figure 10 presents that when the real part of the impedance is constant, the resonance frequency $f_{s}$ and the half power point $f_{1}$ increase as the value of the imaginary part increases. Furthermore, the imaginary part of the impedance is constant, and the dynamic resistance *R* is added as the real part increases.

Further, the experiment was conducted when gradually increasing the weight at the front cover end from 0 N to 40 N, and multiple sets of load characteristic parameter data were measured. Then one set of data is selected arbitrarily, for example, when a force of 20 N is applied, and measured by PV520A impedance analyzer $f_{s}^{e}$ = 20,030.4 Hz, $f_{1}^{e}$ = 19,998.7 Hz and $R^{e}=15.6743\;\Omega$. The values of the load characteristic parameters measured for the selected apply 20 N force are each made in 3D coordinates with a constant level to determine the same values of the study characteristic parameters in the impedance range of 0 to 1000−1000i. Finally, three fixed-value horizontal surfaces are used for the horizontal equivalent cross-section of the 3D.

Three-dimensional plots of load characteristic parameters ($f_{s}$, $f_{1}$, *R*) are intersected with the plane plotted by the values of the load characteristic parameters ($f_{s}^{e}$, $f_{1}^{e}$, $R^{e}$) and measured when applying a force of 20 N to visualize the position of intersection curves in a two-dimensional coordinate system. The cross curves are mapped individually to the two-dimensional plane, which consists of the x-axis (real part of the impedance) and y-axis (imaginary part of the impedance). The mapping process of the intersection curves is shown in Figure 11. The intersection curves are fully mapped to the two-dimensional coordinate system; the three intersection curves are shown in Figure 12.

In order to obtain the corresponding impedance under the load, the intersection lines merge together as shown in Figure 13a. From Figure 13a, we can obtain the three intersection points by three map curves (65,514), (65,472), and (122,529). According to the relevant mathematical knowledge, we make an inscribed circle for three intersection points, as shown in Figure 13b. The center of this circle can be obtained as (78,504); it can be approximated to the impedance corresponding to equal-load characteristic parameters when the load applies to this transducer. Considering the characteristics of equal distance from the center of the circumcircle to the intersection point, we have also made a circumcircle for three intersections, as shown in Figure 13c.

Gradually adding weight to the metal front cover, the load characteristic parameters are measured using a PV520A impedance analyzer for each load. When we repeat these operations (constant cross-section, map intersection curves, merge intersection lines, make an inscribed circle and circumcircle) and the impedance values corresponding to this transducer when applying different loads can be obtained. The impedance and load obtained by different methods are shown in Table 4, Table 5 and Table 6.

### 3.4. An Equivalent Model of Piezoelectric Transducer about the Impedance and Load Is Established

**Remark 3.** 
*Fitting the real part ($Re(Z_{ft})$) of local optimal impedance ($Re\left(Z_{ft}\right)-Im\left(Z_{ft}\right)i$) and force F by polynomial. Simultaneously, we use this method to fit the imaginary part ($Im(Z_{ft})$) and F. Then the parameters of the polynomial are identified.*


According to the physical characteristics of the ultrasonic transducer, impedance shows a certain correlation with the load force. By analyzing the experimental data that was obtained by conducting many experiments, it can be seen that the real and imaginary parts of impedance ($Re\left(Z_{ft}\right)-Im\left(Z_{ft}\right)i$) have a certain numerical relationship with load *F*. This is fit by many methods, such as exponential, polynomial, and Gaussian. Finally, it is found that the polynomial method of curve fitting has an excellent effect. Assuming that the function obtained by the fitting is
(7)$$\left\{\begin{array}{c} Re\left(Z_{ft}\right)=a_{1}\times F^{2}+b_{1}\times F+c_{1}\\ Im\left(Z_{ft}\right)=a_{2}\times F^{2}+b_{2}\times F+c_{2}. \end{array}\right.$$

First, about the real part of impedance obtained by the inscribed circle and load *F*, and intermediate parameters by the least squares can be obtained $a_{1}=-0.00345$, $b_{1}=1.286$, and $c_{1}=44.87$, Figure 14a presents the fitting effect; the expression can be expressed as
(8)$$Re\left(Z_{ft}\right)=-0.00345\times F^{2}+1.286\times F+44.87.$$

Simultaneously, the imaginary part and force-fitting effect in Figure 15a, we can obtain the parameters $a_{2}=-0.08822$, $b_{2}=18.39$, and $c_{2}=139.6$, the functional equation obtained denotes
(9)$$Im\left(Z_{ft}\right)=-0.08822\times F^{2}+18.39\times F+139.6.$$

Similarly, we can fit the impedance obtained by the circumcircle to the added load *F*; this is shown in Figure 14b and Figure 15b. The relationship is
(10)$$\left\{\begin{array}{c} Re\left(Z_{ft}\right)=-0.005796\times F^{2}+1.493\times F+61.58\\ Im\left(Z_{ft}\right)=-0.08322\times F^{2}+18.13\times F+132.7. \end{array}\right.$$

The average impedance of the inscribed circle and circumcircle is fitted to the load *F*, the fitting effect is shown in Figure 14c and Figure 15c. The fitting relationship is
(11)$$\left\{\begin{array}{c} Re\left(Z_{ft}\right)=-0.004451\times F^{2}+1.384\times F+53.12\\ Im\left(Z_{ft}\right)=-0.08569\times F^{2}+18.26\times F+136.1. \end{array}\right.$$

From Figure 14 and Figure 15, we can obtain a fitting effect with regard to the three approaches proposed. By comparing some of the process parameters, we finally determine that the model (Equations (Equation 8) and (Equation 9)) is the best equivalent model. Next, we will verify this model with a large amount of data.

## 4. Experimental Verification

In order to verify the reliability of the built model, we select any transducer from the same batch (the resonant frequency is around 20 KHz) for verification. First, apply a load *F* arbitrarily, according to Equations (Equation 8)–(Equation 11), and an impedance ($Re\left(Z_{ft}\right)-Im\left(Z_{ft}\right)i$) can be calculated, making the range of the applied force to be 0 to 40 N. Then in the MATLAB simulation environment, based on the information of piezoelectric transducer in Table 2 as basic information, Equations (Equation 1)–(Equation 6) are used as a calculation part, and the value of the load characteristic parameter ($f_{s}^{s}$, $f_{1}^{s}$, $R^{s}$) corresponding to several impedance are calculated. At the same time, the load characteristic parameters ($f_{s}^{e}$, $f_{1}^{e}$, $R^{e}$) of the selected transducer when loaded are measured using a PV520A type impedance analyzer. The values of load characteristic parameters ($f_{s}^{e}$, $f_{1}^{e}$, $R^{e}$) and ($f_{s}^{s}$, $f_{1}^{s}$, $R^{s}$) are plotted in a graph following the tracing point method, as shown in Figure 16, Figure 17 and Figure 18.

By comparative analysis of Figure 16, Figure 17 and Figure 18, we can know that the impedance obtained from the inscribed method is closest to the experiment value. The error analysis of the load parameters obtained by the built model and the experimental measurements is carried out, as shown in Table 7 and Table 8. Finally, we can know that the functional relationship Equations (Equation 8) and (Equation 9) are the optimal correspondence between the impedance and load.

After comparison, the experimental and simulation data matched within the error range. It is verified that the proposed cross-value mapping method can achieve a good correspondence between the load and impedance of the transducer. By performing error analysis on the two groups of data $f_{s}^{s}$, $f_{1}^{s}$, $R^{s}$ and $f_{s}^{e}$, $f_{1}^{e}$, $R^{e}$, obtaining the error analysis Table 7 and Table 8. We can learn that the experiment and simulation results are within the error range, and our method is feasible. Combining Figure 16, Figure 17 and Figure 18 with Table 7 and Table 8, we can obtain: (1) in the (15 N–30 N) interval, $f_{1}$ deviates more, and $f_{s}$ deviates less, but the opposite is true in the other ranges. (2) The overall deviation of the dynamic resistance *R* is greater than $f_{s}$ and $f_{1}$.

In the process of ultrasonic welding, the amplitude and stability of the output end of the tool head determine the welding quality of the ultrasonic welding system, so it is very important to obtain a stable output amplitude. Due to the change of temperature, load, etc., the amplitude of the output end of the piezoelectric transducer is unstable, which affects the machining quality and workpiece life. Based on the equivalent model of load and impedance, we know that the change in impedance is estimated according to the change in the load. This leads to knowing the change in amplitude indirectly. And the amplitude fluctuation can be compensated by the changes in load based on the static feedforward control method. This provides the conditions for having controllers designed with good control effects, making the amplitude more stable during ultrasonic welding.

## 5. Conclusions

Load characteristics analysis is necessary during ultrasonic welding, but it has always been a major challenge. Since the load direction in the ultrasonic welding process is mainly along longitude, and the experimental platform built in this paper is consistent with the load direction during real welding and constructs the equivalent model of the load impedance, which can achieve accurate and efficient load dynamic analysis. We analyze the characteristic of the loading test and the front cover plate radiated acoustic ($Z_{ft}$), which paves the way for the later modeling. We propose a cross-value mapping method that maps the changes in different loads to changes in impedance. By making an inscribed circle on the three mapped intersections, we can obtain an impedance corresponding to the load. The impedance and load are analyzed and fitted by polynomials, and parameter identification of polynomials by using the least squares method. Through verification, we found that the model has a good effect. It can be seen that the model we have built is of great significance to achieve high-quality, high-efficiency, low-cost ultrasonic welding. Next, we will further investigate the transducer controller based on the model established in this article.

## Figures and Tables

**Figure 1 sensors-22-08576-f001:**
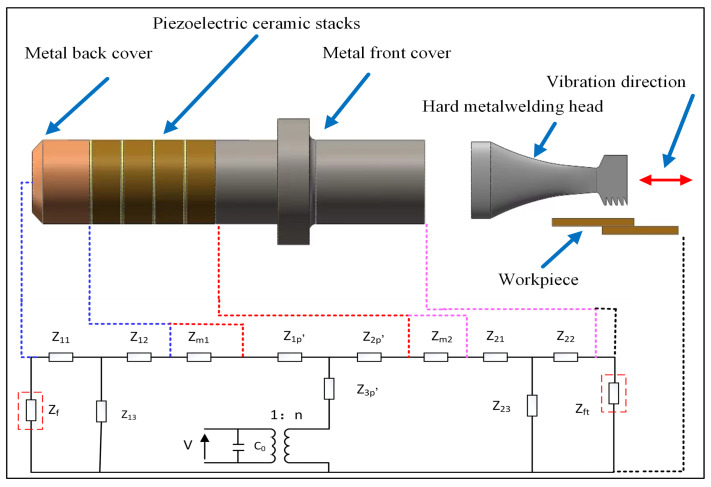
Electromechanical equivalent model of the piezoelectric transducer.

**Figure 2 sensors-22-08576-f002:**
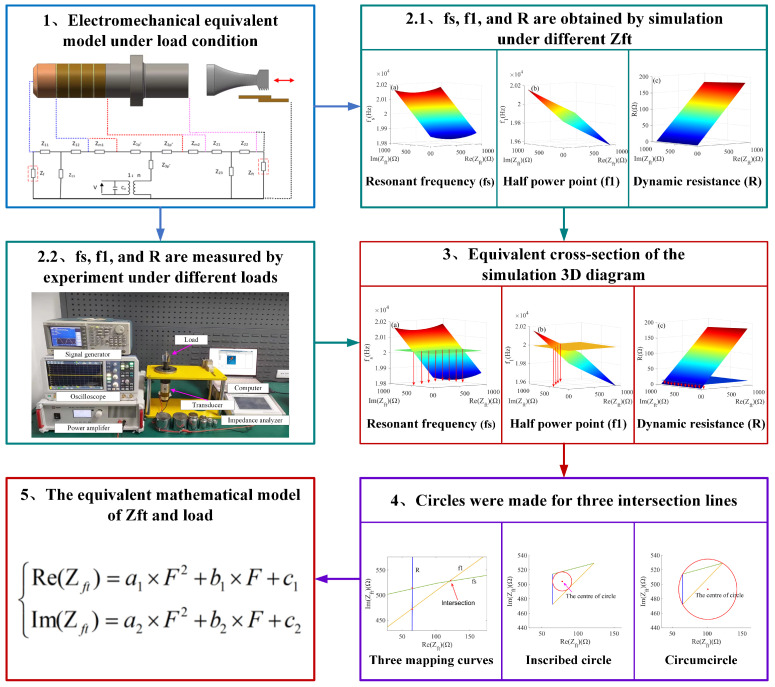
The main research content.

**Figure 3 sensors-22-08576-f003:**
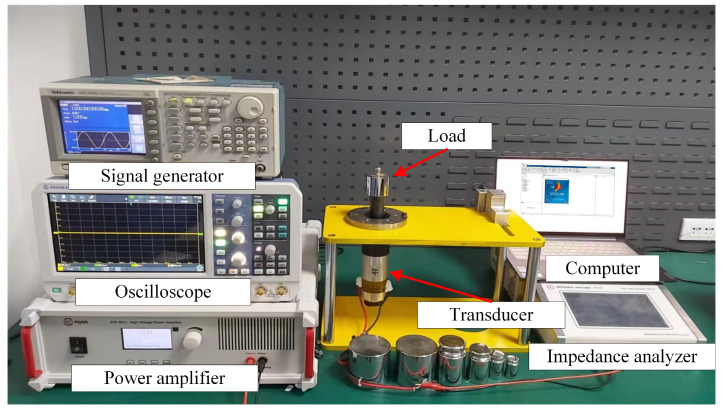
The experiment platform.

**Figure 4 sensors-22-08576-f004:**
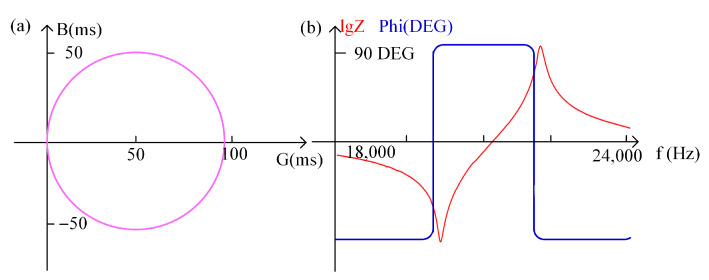
(**a**) Piezoelectric transducer impedance analysis admittance circle; (**b**) Piezoelectric transducer impedance analysis amplitude phase curve.

**Figure 5 sensors-22-08576-f005:**
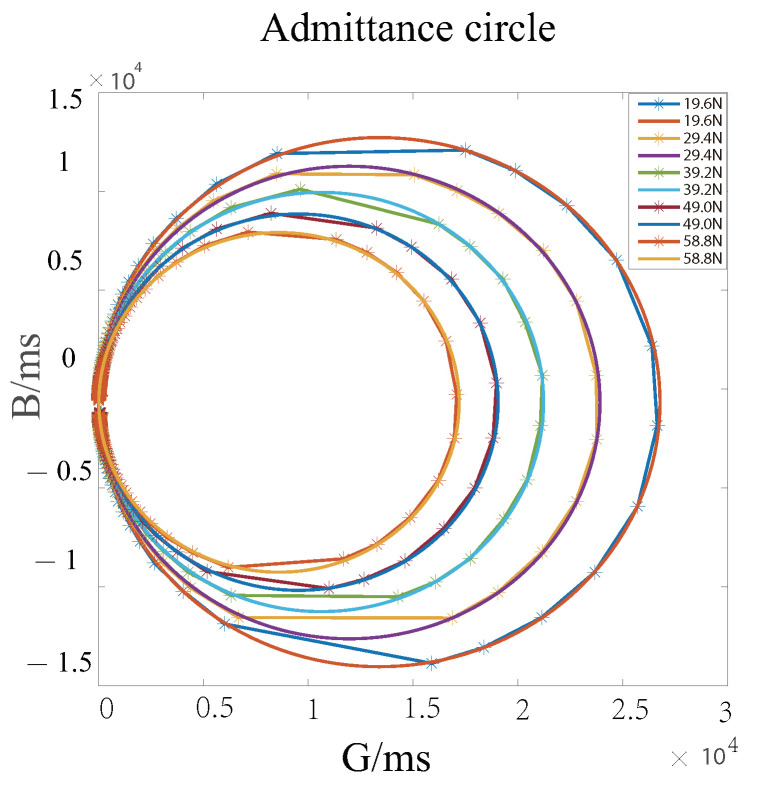
Admittance circle of the piezoelectric transducer under different load.

**Figure 6 sensors-22-08576-f006:**
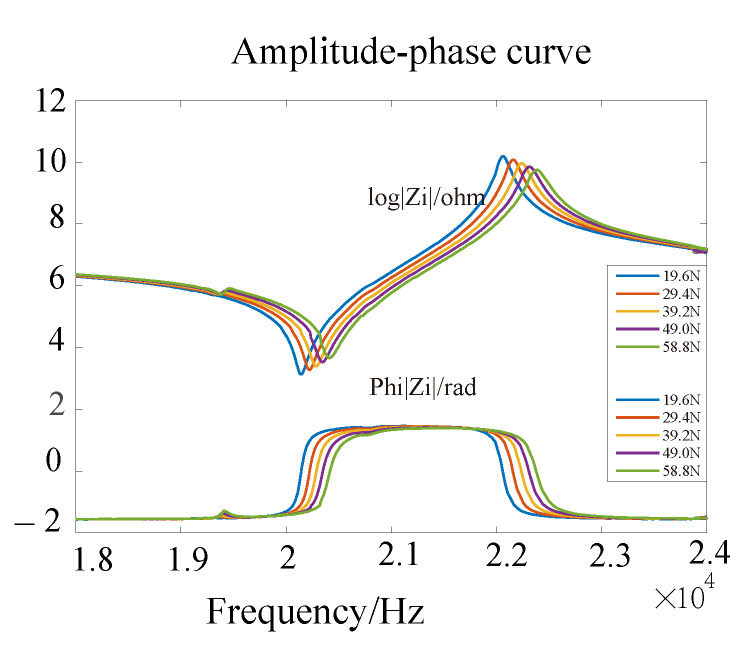
Amplitude-phase curve of the piezoelectric transducer under different load.

**Figure 7 sensors-22-08576-f007:**
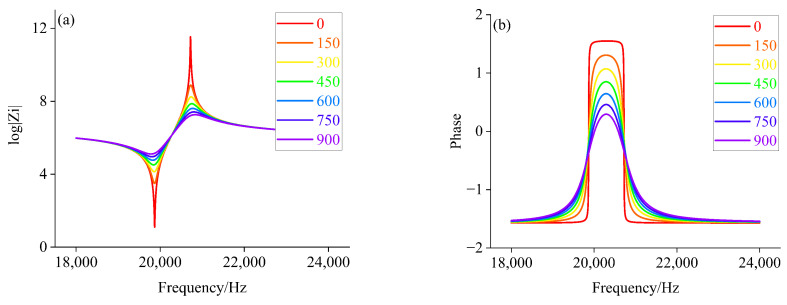
(**a**) Amplitude value characteristics when $Z_{ft}$ is a resistive load; (**b**) Phase frequency characteristics when $Z_{ft}$ is a resistive load.

**Figure 8 sensors-22-08576-f008:**
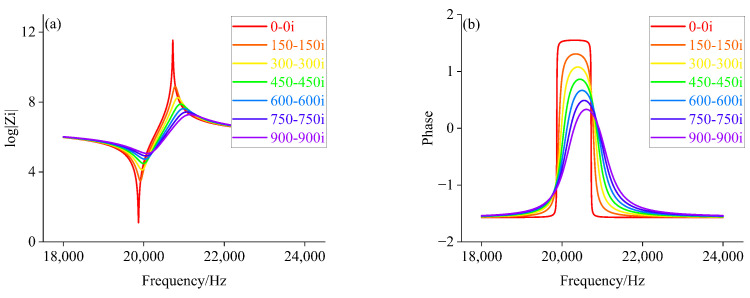
(**a**) Amplitude value characteristics when $Z_{ft}$ is a capacitive load; (**b**) Phase frequency characteristics when $Z_{ft}$ is a capacitive load.

**Figure 9 sensors-22-08576-f009:**
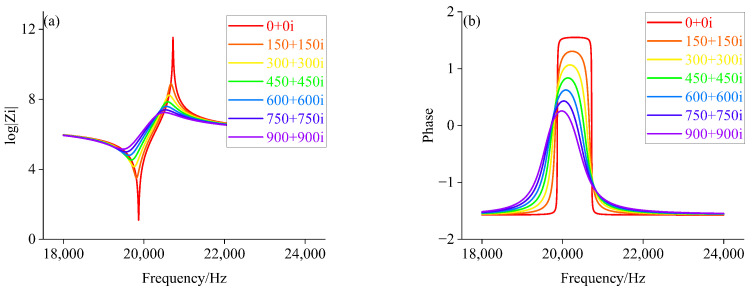
(**a**) Amplitude value characteristics when $Z_{ft}$ is a inductive load; (**b**) Phase frequency characteristics when $Z_{ft}$ is a inductive load.

**Figure 10 sensors-22-08576-f010:**
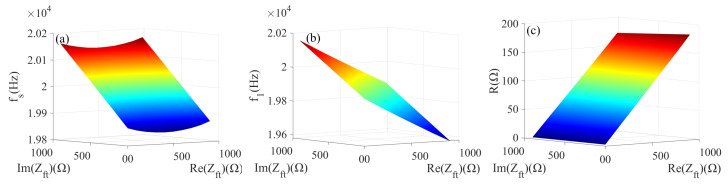
(**a**) $f_{s}$ and $Z_{ft}$; (**b**) $f_{1}$ and $Z_{ft}$; (**c**) *R* and $Z_{ft}$.

**Figure 11 sensors-22-08576-f011:**
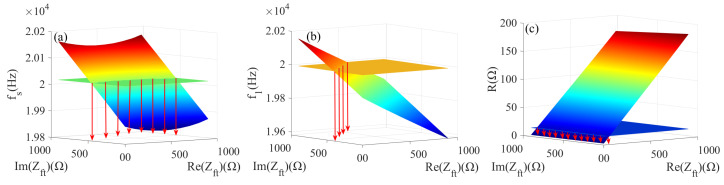
(**a**) The intersection mapping of $f_{s}$; (**b**) The intersection mapping of $f_{1}$; (**c**) The intersection mapping of *R*.

**Figure 12 sensors-22-08576-f012:**
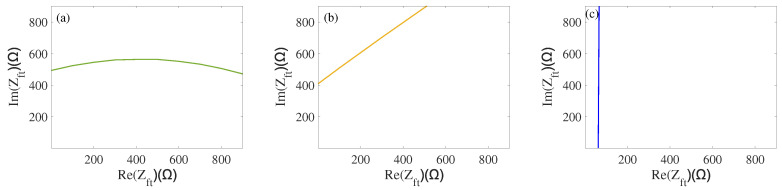
(**a**) Two-dimensional intersection of $f_{s}$; (**b**) Two-dimensional intersection of $f_{1}$; (**c**) Two-dimensional intersection of *R*.

**Figure 13 sensors-22-08576-f013:**
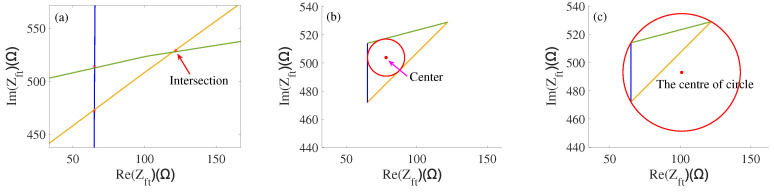
(**a**) Mapping curve intersection diagram; (**b**) Inscribed circle diagram; (**c**) Circumcircle diagram.

**Figure 14 sensors-22-08576-f014:**
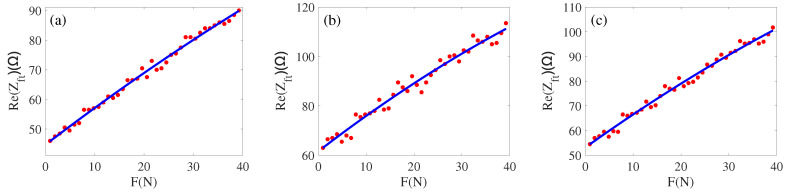
(**a**) Load and real part by an inscribed circle; (**b**) Load and real part by a circumcircle; (**c**) Load and real part by average.

**Figure 15 sensors-22-08576-f015:**
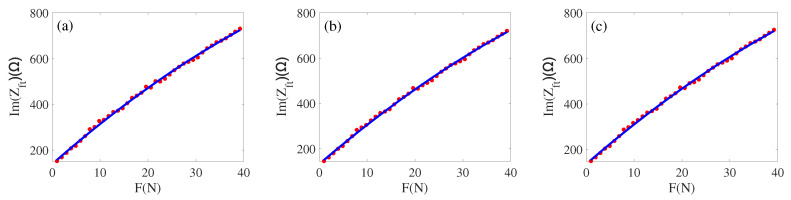
(**a**) Load and imaginary part by an inscribed circle; (**b**) Load and imaginary part by a circumcircle; (**c**) Load and imaginary part by average.

**Figure 16 sensors-22-08576-f016:**
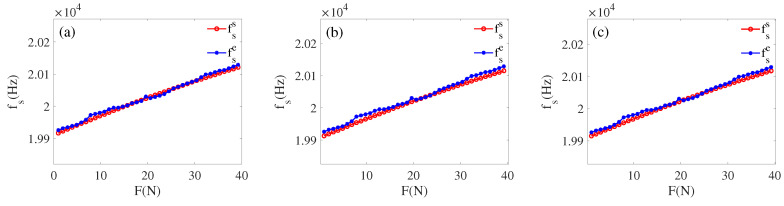
(**a**) $f_{s}^{e}$ and $f_{s}^{s}$ by inscribed circle; (**b**) $f_{s}^{e}$ and $f_{s}^{s}$ by circumcircle; (**c**) $f_{s}^{e}$ and $f_{s}^{s}$ by average.

**Figure 17 sensors-22-08576-f017:**
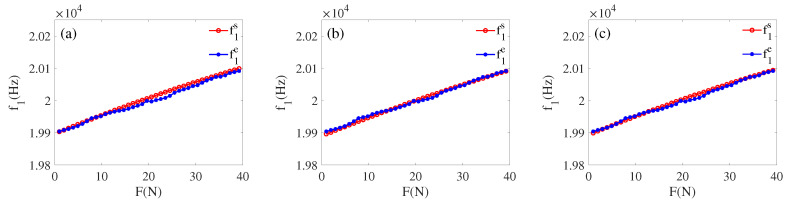
(**a**) $f_{1}^{e}$ and $f_{1}^{s}$ by inscribed circle; (**b**) $f_{1}^{e}$ and $f_{1}^{s}$ by circumcircle; (**c**) $f_{1}^{e}$ and $f_{1}^{s}$ by average.

**Figure 18 sensors-22-08576-f018:**
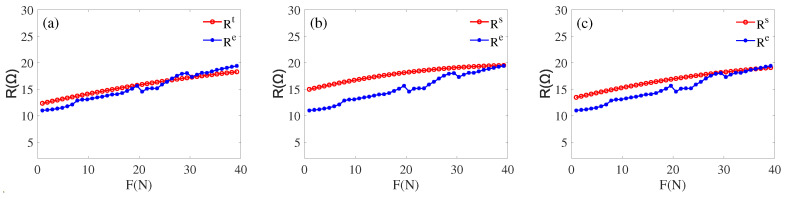
(**a**) $R^{e}$ and $R^{s}$ by inscribed circle; (**b**) $R^{e}$ and $R^{s}$ by circumcircle; (**c**) $R^{e}$ and $R^{s}$ by average.

**Table 1 sensors-22-08576-t001:** Simplification of parameters.

Name of Variable	Variable Symbol
Simulation $f_{s}$	$f_{s}^{s}$
Experiment $f_{s}$	$f_{s}^{e}$
Simulation $f_{1}$	$f_{1}^{s}$
Experiment $f_{1}$	$f_{1}^{e}$
Simulation *R*	$R^{s}$
Experiment *R*	$R^{e}$
Real part of impedance	$Re(Z_{ft})$
Imaginary part of impedance	$Im(Z_{ft})$
Pressure load	*F*

*f_s_* represents forward resonant frequency. *f*_1_ represents half power point. *R* represents dynamic resistance

**Table 2 sensors-22-08576-t002:** The impedance of different components.

Component	Parameters				The Replaced Parameters	
Ceramic	$C_{p}=S_{p}v_{p}\rho_{p}$	$\alpha_{p}=mk_{p}l_{p}$	$S_{p}=\frac{C_{p}}{\sin\alpha_{p}}$	$T_{p}=C_{p}\tan\frac{\alpha_{p}}{2}$	$Z_{1p^{\prime }}=Z_{2p^{\prime }}=jT_{p}$	$Z_{3p^{\prime }}=jS_{p^{\prime }}$
Rear cover	$C_{1}=S_{1}v_{1}\rho_{1}$	$\alpha_{1}=mk_{1}l_{1}$	$S_{1}=\frac{C_{1}}{\sin\alpha_{1}}$	$T_{1}=C_{1}\tan\frac{\alpha_{1}}{2}$	$Z_{11}=Z_{12}=jT_{1}$	$Z_{13}=jS_{1}$
Front cover	$C_{2}=S_{2}v_{2}\rho_{2}$	$\alpha_{2}=mk_{2}l_{2}$	$S_{2}=\frac{C_{2}}{\sin\alpha_{2}}$	$T_{2}=C_{2}\tan\frac{\alpha_{2}}{2}$	$Z_{21}=Z_{22}=jT_{2}$	$Z_{23}=jS_{2}$

**Table 3 sensors-22-08576-t003:** Relevant parameters information of the piezoelectric transducer.

Component	Attributes	Value
Ceramic	Material	PZT-4
	Density (kg/m^3^)	7600
	Number of chips	4
	Piezoelectric constant	$2.18\times 10^{-10}$
Rear cover	Material	Aluminum, Steel
	Density (kg/m^3^)	4370
	Young’s modulus (N/m^2^)	$7.23\times 10^{10}$
	Length (mm)	32
Front cover	Material	Rigid aluminum
	Density (kg/m^3^)	2700
	Young’s modulus (N/m^2^)	$7.15\times 10^{10}$
	Length (mm)	54

**Table 4 sensors-22-08576-t004:** The impedance obtained by the inscribed circle.

F(N)	0.98	1.96	2.94	3.92	4.90	…	37.24	38.22	39.20	…
$Re\left(Z_{ft}\right)\;\left(\Omega \right)$	46	48	49	50	50	…	87	89	90	…
$Im\left(Z_{ft}\right)\;\left(\Omega \right)$	152	169	188	208	219	…	704	718	731	…

**Table 5 sensors-22-08576-t005:** The impedance obtained by the circumcircle.

F(N)	0.98	1.96	2.94	3.92	4.90	…	37.24	38.22	39.20	…
$Re\left(Z_{ft}\right)\;\left(\Omega \right)$	63	67	67	69	66	…	106	110	114	…
$Im\left(Z_{ft}\right)\;\left(\Omega \right)$	146	162	181	200	213	…	695	707	721	…

**Table 6 sensors-22-08576-t006:** The impedance obtained by the average of the circumcircle and the inscribed circle.

F(N)	0.98	1.96	2.94	3.92	4.90	…	37.24	38.22	39.20	…
$Re\left(Z_{ft}\right)\;\left(\Omega \right)$	55	57	58	60	58	…	96	99	102	…
$Im\left(Z_{ft}\right)\;\left(\Omega \right)$	149	166	184	204	216	…	699	712	726	…

**Table 7 sensors-22-08576-t007:** 0–30 N error analysis.

F(N)	16.66	17.64	18.62	19.60	24.50	25.48	26.46	27.44	28.42	29.40
$f_{s}$ (Hz)	0.01%	0.01%	0.01%	0.03%	0.02%	0.00%	0.00%	0.01%	0.01%	0.01%
$f_{1}$ (Hz)	0.06%	0.06%	0.06%	0.04%	0.08%	0.06%	0.06%	0.06%	0.06%	0.05%
R(Ω)	6.56%	4.89%	3.16%	0.62%	3.63%	1.27%	1.60%	3.94%	5.28%	5.26%

**Table 8 sensors-22-08576-t008:** 30–40 N error analysis.

F(N)	30.38	31.36	32.34	33.32	34.30	35.28	36.26	37.24	38.22	39.20
$f_{s}$ (Hz)	0.00%	0.03%	0.05%	0.04%	0.04%	0.04%	0.02%	0.03%	0.04%	0.04%
$f_{1}$ (Hz)	0.06%	0.04%	0.03%	0.03%	0.02%	0.04%	0.04%	0.03%	0.03%	0.04%
R(Ω)	0.23%	2.12%	3.34%	2.63%	3.51%	4.41%	4.83%	5.29%	5.82%	6.10%

## Data Availability

Not applicable.

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
