# Peer review of "Analysis of Ultrasonic Machining Characteristics under Dynamic Load"

_sensors, 2022, doi:10.3390/s22218576_

Round 1
Reviewer 1 Report
The paper investigates the impedance characteristics of the piezoelectric transducer during loading. It is a topic of interest to the researchers in the areas but the paper needs some improvement before acceptance for publication. Some detailed comments are as follows:
1. In the introduction, the author should explain the difference from other scholars' research and elaborate on the focus of this work. In addition, there are more relevant papers that should be covered in literature review:
https://doi.org/10.1016/j.ijmachtools.2020.103594
https://doi.org/10.1016/j.ijmecsci.2022.107375
2. During ultrasonic-assisted machining, changes in load may cause the stability of the amplitude. How to take this into account in your design?
3. In conclusions:
Author should highlight the conclusion over here. Therefore, it is necessary to modify this to make the expression clearer.
Reviewer 2 Report
please find my comments in the attached pdf

Reviewer 3 Report
Comments:
for a "small" review of literature
no links in the text to equations (some are)
missing all units in Figures 10-14
the title does not exactly correspond to the content presented in the article (maybe not vibration, but dynamic analysis)
Round 2
Reviewer 2 Report
Dear Authors,
thank you for carefully revising the manuscript. I still doubt that the mass load is well suited to mimick the welding process. As the welding process consumes energy, is should add damping characteristics to the system. A mass doesn't do that.
Further on, what will happen, if the vibration shape is influenced by the load? Ok, you might neglect this for your special application, but if so, please write that before all your calculations.
Another thing, you did not change: "... the impedance characteristic ... is capacitive." -> The characteristics change frequency-dependent between capacitive, inductive and resistive. Thus you have to mention the frequency or frequency band which you are interested in!
A pure resistive load is defined by a pure real value of Z. A pure capacitive or inductive load by a pure imaginary value of Z. A value of (1 + 1i) or similar is not a pure, but a mixed load.
I do not agree to your last sentences in chapter 4: based on your model you could estimate the change in current, but thatfor you need the load characteristics during the process, which you don't have.
I'm eager to see a controller working based on your results...
